# The Structural Types of the Polarization Detection Unit in Imaging Polarimeter Based on the Stokes Parameter Method

**DOI:** 10.3390/s25134069

**Published:** 2025-06-30

**Authors:** Yuanhao Li, Xiaohan Guo, Kai Zhang, Xiaoyang Li, Fang Kong, Ziying Jia

**Affiliations:** 1College of Underwater Acoustic Engineering, Harbin Engineering University, Qingdao 266000, China; lilyeh@hrbeu.edu.cn (Y.L.); kaizhang@hrbeu.edu.en (K.Z.); lixiaoyang@hrbeu.edu.cn (X.L.); jzying@hrbeu.edu.cn (Z.J.); 2College of Electrical Engineering and Automation, Shandong University of Science and Technology, Qingdao 266590, China; 202181080004@sdust.edu.cn

**Keywords:** polarization imaging, bionic navigation, Stokes parameter, sensor structure

## Abstract

Bio-inspired imaging polarimeters have significant applications in the field of detecting the polarization state of skylights. The polarization detection principle of polarization detection units in polarimeters is mostly based on the Stokes parameter method. Using the Stokes parameter method, multiple linearly polarized lights modulated by the incident light need to be obtained. According to the polarization modulation method of the polarization detection unit, imaging polarimeters can be classified into time-division types, channel-division types, and division of focal-plane types. Different from the classification in previous studies, this review divides channel-division polarimeters into single-sensor channel-division and multi-sensor channel-division polarimeters, avoiding the confusion of concepts between aperture-sharing polarimeters and amplitude-sharing polarimeters in previous classifications. This review introduces the different ways of achieving polarization-state imaging through various bionic imaging polarimeters and expands on the advanced polarization detection unit structure design technologies based on the Stokes parameter method introduced in recent years, aiming to provide inspiration for bio-inspired imaging polarimeters used in navigation and positioning.

## 1. Introduce

Polarization-state detection technology plays a significant role in multiple fields, such as navigation, detection, and enhanced imaging. For instance, detecting the I have updated some references during the revision. To prevent conflicts between EndNote and the Microsoft Word’s built-in track changes function, I disabled the track changes function while adjusting the references. I can guarantee that no information in the original version has been altered. Please accept these changes.polarization state of skylight can obtain abundant directional information in the skylight, thereby achieving positioning and navigation [1,2]. Detecting the polarization state of reflected light from substances and laser light can obtain polarization information that traditional light intensity detection technology does not possess, enhancing related images and enabling photoelectric detection technology to play a better role in biomedicine, material detection, and other aspects [3,4]. The Stokes vector method is the main computational method for obtaining polarization information of light, and the research on polarimeters based on the Stokes vector method is one of the research hotspots in the field of polarization detection unit structure design [5,6]. Bionic imaging polarimeters are developed based on the mechanism of animal eyes. They use lenses and imaging sensors to measure the polarization patterns of the entire sky within a wide field of view and extract polarization information by analyzing the intensity differences between different images. The polarization detection unit structures of bionic imaging polarimeters are mostly designed based on the Stokes parameter method, according to different principles, and they can be classified into time-division types, channel-division types, and division of focal-plane types.

Inspired by the fact that certain species can utilize the polarization information of light to obtain direction [7,8], currently, the bionic polarization imaging polarimeters are mainly designed to obtain and process the polarization information of skylight, acquire the direction related to the sun [9], and lay the foundation for subsequent navigation and positioning [10], or they can be combined with various navigation methods and work as auxiliaries for navigation and positioning [11,12,13]. In fact, the application of polarimeters based on the Stokes vector method is very extensive; it includes material detection [14] and polarization state verification [15], and the research on bionic polarization skylight is not limited to navigation [16]. This paper focuses on bionic imaging polarimeters for navigation and positioning purposes, based on the analysis of polarization state detection principles in various bionic imaging polarimeters. We also cite the development trend of polarization detection unit structure design, aiming to provide inspiration for bionic imaging polarimeters for navigation and positioning purposes.

## 2. The Optical Polarization State Detection Principle Based on the Stokes Parameters Method

According to whether the electric field intensity vector axis rotates around the optical axis, polarized light can be classified into linearly polarized light, circularly polarized light, elliptically polarized light, etc.; specifically, circularly polarized light can be further categorized as left-handed circularly polarized (LHCP) or right-handed circularly polarized (RHCP) depending on the clockwise or counterclockwise rotation direction of the electric field vector, as shown in Figure 1 [17]. Scattered by particles in the atmosphere, the unpolarized sunlight is modulated into polarized light and forms an ideal and stable polarization distribution pattern on a macroscopic scale, which contains rich positional information such as the meridian, degree of polarization (DOP), and angle of polarization (AOP). By observing the skylight polarization pattern, one can obtain positional information between the observation point and the Sun for navigation purposes.

### 2.1. Stokes Vector and Mueller Matrix Overview

The light intensity information is mathematically processed through the Stokes–Mueller matrix to describe the polarization state of light, where the light can be monochromatic, non-monochromatic, fully polarized, partially polarized, or natural light. The four Stokes parameters are the time-averaged values of the light intensity. Assuming that the light propagates along the *z*-axis, we then have the following:(1)$$\left[S\right]=\left[\begin{array}{c} S_{0}\\ S_{1}\\ S_{2}\\ S_{3} \end{array}\right]=\left[\begin{array}{c} E_{x}^{2}+E_{y}^{2}\\ E_{x}^{2}-E_{y}^{2}\\ 2E_{x}E_{y}cos\delta \\ 2E_{x}E_{y}sin\delta \end{array}\right]$$
where $E_{x}$ represents the time-averaged value of the amplitude of the light in the x-direction, $E_{y}$ represents the time-averaged value of the amplitude of the light in the y-direction, and δ denotes the phase difference of the light in the x-axis and y-axis directions. The terms $S_{0}$ to $S_{3}$ are called Stokes parameters, with each parameter describing a specific characteristic of the polarization of the light. $S_{0}$ describes the total intensity, $S_{1}$ describes the degree of polarization in the vertical or horizontal direction, $S_{2}$ describes the degree of polarization at 45° to the *x*-axis or *y*-axis in the propagation direction, and $S_{3}$ describes the circular polarization characteristics.

The principle of the Stokes vector method for measuring light polarization can be summarized as follows: any polarization state of light can be represented by four measurable parameters. The subsequent derivation of specific measurement methodologies and calculation formulas for Stokes parameters can be attributed to the Mueller matrix formalism. The Mueller matrix provides a complete mathematical description of the polarization characteristics when objects interacting with light. It is a transfer function of the Stokes vector, used to characterize the polarization state of light and describe the polarization properties of media [18]. Here, the Mueller matrix describes the interaction between linear polarizers and polarized light.

### 2.2. Four-Angle Measurement Method

The polarization detection unit of a polarimeter transforms incident light with an unknown polarization state into multiple known polarization states. To determine all four Stokes parameters, four different measurements are supposed to be performed by using linearly polarized filters, as follows:(2)$$\left[S\right]=\left[\begin{array}{c} E_{x}^{2}+E_{y}^{2}\\ E_{x}^{2}-E_{y}^{2}\\ 2E_{x}E_{y}cos\delta \\ 2E_{x}E_{y}sin\delta \end{array}\right]=\left[\begin{array}{c} I\\ Q\\ U\\ V \end{array}\right]=\left[\begin{array}{c} I_{0^{o}}+I_{90^{o}}\\ I_{0^{o}}-I_{90^{o}}\\ I_{45^{o}}-I_{135^{o}}\\ I_{R}-I_{L} \end{array}\right]$$
where $I_{i^{o}}$ represents the intensity of the light passing through a polarizer at an angle of $i^{o}$, and $I_{R}$ and $I_{L}$ represent the intensities of right-handed and left-handed circularly polarized light, respectively. After obtaining the four Stokes parameters, various polarization-attribute information of the light can be calculated, namely the angle of polarization (AOP), degree of polarization (DOP), degree of linear polarization (DOLP), and degree of circular polarization (DOCP). The formulas are as follows [19]:(3)$$DOP=\frac{\sqrt{Q^{2}+U^{2}+V^{2}}}{I}$$(4)$$DOLP=\frac{\sqrt{Q^{2}+U^{2}}}{I}$$(5)$$DOCP=\frac{\sqrt{V^{2}}}{I}$$(6)$$AOP=\frac{1}{2}arctan\left[\frac{U}{Q}\right]$$

### 2.3. Three-Angle Measurement Method

It is worth noting that polarized skylight is mainly linear polarized light, and there is little circular polarized light; it is considered that V = 0. Thus, if only DOP and AOP information is required, there are three unknowns that need to be solved, meaning that only three equations are required. Therefore, the polarization light can be expressed in another form [20]. If the light passes through the polarizer at an angle of α relative to the x-axis, the intensity of the light can be expressed as follows:(7)$$I_{\alpha }=\frac{1}{2}(I+Qcos\left(2\alpha \right)+Usin(2\alpha ))$$

Equation (7) is applicable exclusively under the condition where circularly polarized light is disregarded, in which case, we only need to measure at just three polarizer angles. From this, it can be known that the installation angles of polarizers in imaging polarimeters are commonly set to 0°, 30°, 45°, 60°, 90°, 120°, 135°, etc., because the polarization degree and polarization angle can be more easily calculated.

The method and principle of obtaining multiple polarization states depend on the structure of the detection unit. In fact, the acquisition of Equation (7) benefits from the Mueller matrix form. Depending on the structure and principle of the polarimeter detection unit, the form of the Mueller matrix and the final calculation formula may vary. However, regardless of the structure, multiple polarization intensity images must be obtained.

## 3. Design of Polarization Detection Unit Structure for Bionic Imaging Polarimeter

Early bionic imaging polarimeters often employed point-source structures to obtain information from a single pixel in the sky [21,22]. The advantage of the point-source polarimeter is that it mimics the vertical structure of biological photoreceptor organs, equipped with a pair of photodetectors that are sensitive to linearly polarized light, forming an antagonistic effect on the acquired light signals [23,24]. Point-source polarimeters perform well in stationary, simple environments, but the pixel information they capture is limited. In complex environments, the imaging performance becomes unstable compared to imaging polarization sensors that can acquire information over a wide field of view [25]. Imaging polarimeters, equipped with lenses that capture a wide field of view, address the instability under complex dynamic conditions. A typical bionic imaging polarimeter consists of the wide-angle lens, polarizers, and an image sensor chip. Specifically, fisheye lenses, known for their exceptionally wide viewing angles, are widely used [26]. The polarizer is generally made of metal grids, which are crucial devices that convert unpolarized light into linearly polarized light, facilitating subsequent polarization state calculations [18]. The image sensor chip is usually composed of a charge-coupled device (CCD) or complementary metal–oxide–semiconductor (CMOS). These components are assembled in various configurations to form the polarization detection unit. The design principles of different configurations are all based on the Stokes vector method [27], acquiring polarization information from multiple polarization states.

As discussed in Section 2, the key in the design of a bionic imaging polarimeters’ detection unit is to employ a reasonable structure that can obtain various polarization intensity images corresponding to different polarization states. Based on different implementation methods, imaging biomimetic polarimeters can be classified into time-division, channel-division, and focal-plane-division categories. The basic imaging principle for all these types involves a wide-angle lens that collects polarized light from the sky into the field of view of the polarization detection unit. After passing through optical lenses, the polarized light reaches the polarizer, which filters out light perpendicular to the direction of propagation, leaving only light parallel to the transmission direction, thereby providing parameters for Stokes vector calculation and allowing the polarization calculation unit to compute the intensity of the linearly polarized light [2].

The traditional time-division polarimeter requires only a single camera, capturing images when polarizer is rotated to distinct orientations [28]. This method relies on rotating the polarizer over time to acquire multiple polarization images, taking a certain amount of time, sacrificing the real-time performance of measurements, and making it suitable for static scene polarization measurements. Further, the electrically controlled time-division polarimeter, which uses electrical control technology to achieve phase delay and measure varied Stokes parameters, does not require rotating mechanical components. This results in stronger stability and dynamic measurement capabilities [29]. It is worth noting that advanced electrical control delay technology is mostly applied to the construction of retarders and the detection of laser polarization states; its application to skylight polarization identification for navigation and positioning is still limited. This offers insights and directions for future design improvements in time-division polarimeters. Channel-division polarimeters use different sensors [30] or different regions of a sensor to capture images [31]. Unlike previous classifications, this review divides channel-division polarization imaging sensors based on the implementation method into those that use multiple sensors and those that use a single sensor. The former uses a series of polarization imaging sensor systems, each equipped with a polarizer assembled in specific direction, to capture and process polarization information in various states [25]. The latter uses a single polarization imaging sensor system, but the system is equipped with a polarizer composed of a micro-waveplate array and fixed linear polarizers [32,33]. By positioned polarizers in front of the lens and utilizing the focusing function of the lens, the imaging sensor is divided into diverse parts, each capturing information from different polarization states [34,35,36]. This classification resolves the confusion in previous categorizations that lacked clear distinctions between amplitude-division and aperture-division concepts. The polarization imaging principle of division-of-focal-plane polarimeter (DOFP) is based on micro-polarization arrays. In this configuration, the polarizer is pixelated and then aligned with the sensor pixel array for installation. The overall pixelized polarizer array can be divided into varied cell arrays, with each cell containing 2 × 2 sub-polarizer units, each equipped with a polarizer oriented in a specific direction. This setup allows the four adjacent sub-units in a cell to transmit linearly polarized light in different directions. The final polarization intensity images in multiple polarization states can be obtained through interpolation [37,38,39]. Since the micro-polarization array is fabricated based on nanometer-scale gratings [18,40], and as nanofabrication technologies advance, DOFP is becoming increasingly mainstream. At the same time, the maturation of ultraviolet nanoimprint lithography (UV-NIL) and electron-beam lithography (EBL) techniques has made it easier to integrate the polarizer array and sensor pixel array [41,42], greatly alleviating the issue of light crosstalk between different pixels [43]. Thanks to its small size, good compatibility, ability to simultaneously image, and suitability for dynamic scenes, the DOFP polarimeter has gradually become the mainstream technology in the field of polarized skylight measurement.

### 3.1. Time-Division Polarimeter

As previously mentioned, time-division polarimeters meet the requirement of acquiring multiple Stokes parameters from the time dimension, which offers an advantage for static measurements [44]. There are generally two methods for implementing time-domain separation: rotating polarizer and electrically controlled phase delay [45]. The rotating polarizer method uses physical rotation to dynamically change the polarization angle of the incident light. Based on the modulation method, it has rotating polarizers or rotating retarders [46,47]. It is worth noting that both the rotating retarder method and the electrical control phase delay method described below are still based on the Stokes vector–Mueller matrix, what makes it special is that they incorporate the Mueller matrix of the retarder and detect the changed polarization state of the linear polarization throughout various phases [48]. The mainstream electrical control phase delay methods can be categorized into those based on nematic liquid crystals (NLCs), ferroelectric liquid crystals (FLCs), and photoelastic modulators (PEMs). Both NLCs and FLCs rely on the electro-optic effect of liquid crystals to modulate the phase of light, with the primary difference being their liquid crystal structures and alignment methods [49,50]. The PEM, on the other hand, is a polarization modulator that operates at the resonant frequency of its optical element. When a linear polarized light beam passes through the PEM at its resonant frequency, a sine-phase difference or phase delay is generated between the two orthogonal linear polarization components of the beam [51].

#### 3.1.1. Rotating Polarization Device Technology

In early studies, polarizers were generally assembled directly in front of the camera and manually rotated. In 1997, Voss et al. [52] applied a time-division polarimeter to measure vector information of skylight, designing a polarization radiation distribution camera system based on a fisheye lens, a CCD camera, and a dichroic polarizer integrated into the camera system. By rotating the linear polarizer, polarization images were captured at 0°, 45°, and 90° polarization angles. In 2014, YJ Wang et al. [28] optimized the measurement setup to improve the accuracy of skylight polarization measurements. They established Cartesian coordinate systems for the pixel coordinates of the skylight (p), the lens coordinates (l), and the incident light coordinates (i). During data acquisition, three images were captured with the polarizer oriented in three directions, and the polarization parameters were determined through three exposures. After establishing the algorithm, they obtained the polarizer’s polarizing film axis angle, β=π/3, which is the optimal value of *β* that minimizes the estimation error of the polarization angle Φ. This finding also explains why the fields of view of some insects’ polarization opposition units (POL-OP units) are concentrated around the zenith at an approximate 60° angle. In 2018, L Guan [44] used a rotating polarizer-based polarization measurement system to study the polarization pattern of the sea sky. It was proven that the polarization distribution of the sea sky is basically the same as that of the land, and the effectiveness of the time-sharing polarization imaging system was also verified. This kind of time-sharing polarimeter system has a relatively simple design and is easy to set up, so it is quite attractive; the physical system employed is schematically illustrated in Figure 2. However, a significant drawback is that both the scene and the measurement platform must remain stationary during the measurement process to avoid introducing inter-frame motion.

The manual rotation method of polarizing devices has the drawback of not achieving an adequate frame rate. Replacing manual rotation with a motor-driven polarizer can effectively reduce the time interval between two consecutive frames. In 2018, Y Han et al. [53] improved the imaging method of continuously rotating polarizers by using a rapidly and uniformly rotating polarizer combined with feedback exposure control, along with an improved pipeline-based polarization image processing approach, which greatly enhanced the speed of polarization pattern detection, as shown in Figure 3a. In 2022, N Gu et al. [54,55] invented a polarization modulator consisting of a high-speed brushless DC motor (BLDCM), a high-precision micro motor, and a high-extinction-ratio polarizer. The polarizer is installed with a precise magnetic encoder (ME) on the high-speed BLDCM. The photodetector is integrated with the BLDCM to collect the polarized intensity modulated by the polarizer, as shown in Figure 3c. In 2024, a new method was proposed to precisely calibrate the transmission angle of the polarizer by using a Wollaston prism online, allowing the invented rotating motor to start rotating from any initial angle, thus providing a solid reference for polarization imaging and polarization calibration. In 2024, F. Kong et al. [56] used 3D-printing technology to create mechanical rotating components and employed a stepper motor to drive the polarizer, allowing the polarizer to rotate rapidly in front of the camera lens. They designed the “PSC-004” Polarization Sky Window Compass based on mechanical rotation. This innovative design enables precise and fast rotation of the polarizer, which is essential for dynamic polarization imaging and accurate measurements. The use of 3D printing offers the advantage of customizable and efficient manufacturing of the mechanical components, enhancing the system’s functionality and ease of use, as shown in Figure 3b. Also in 2024, X. Lu et al. [57] proposed a rotation system that uses an encoder to capture the current position and transmit it to the drive circuit board. The drive circuit board generates voltage signals based on the current position to drive the rotation. To address the error accumulation caused by continuous rotation of the polarizer in the polarization rotation stage, they proposed a method to determine the true rotation angle of the polarizer when the rotation angle is unknown, as shown in Figure 3d. All the aforementioned polarimeters for navigation and positioning purposes are designed for direct detection of skylight polarization. In 2023, CQ Zhang [58] proposed a novel portable indirect polarization parameter microscopic imaging system. This system places a polarization-sensitive marking device on the surface of the object being tested. The camera continuously captures images of the object, while the polarizer automatically rotates to modulate the polarization imaging; this allows the system to intelligently measure the polarization information of incident light with unknown polarization angles within the coverage area, providing a new approach for detecting the polarization state of skylight.

Compared to the rotating polarizer method, using a rotating retarder with a fixed polarizer enables full-Stokes parameter imaging. Building upon previous work, the rotating polarizer was replaced with a rotating retarder, while the polarizer was fixed. In 2022, N. Gu [59] proposed a polarization imaging device based on continuous rotation of the retarder and a fixed polarizer, optimizing the recovery and structure of the full-Stokes parameters. This system enabled the full-Stokes vector to be obtained through a single matrix calculation, as shown in Figure 4a. In 2024, JP Yang et al. [48] used two standard retarders with different linear retardance values as polarization modulation units. This setup achieved various special delays with errors within 0.2° across different wavelengths, along with polarization measurement accuracy below the 10^−2^ level, as shown in Figure 4b. However, the measurement speed remains limited, which is a common issue with time-division imaging polarimeters.

#### 3.1.2. Electrically Controlled Phase Delay Technology

The liquid crystal variable retarder (LCVR) can modulate the phase delay of incident light by controlling the applied voltage; due to its ability to electronically adjust the phase of light, LCVR has been widely used in the field of polarization measurements [60,61]. In 2018, H.J. Zhao et al. [62,63] used a ground-based all-sky imaging polarimeter based on LCVR to capture polarization images of the skylight. This system comprises two LCVRs and their respective polarizers, and the LCVR voltage is set to four different values, resulting in four different phase delays. Subsequently, four intensity images corresponding to these phase delays are acquired, and from them, four Stokes parameters are obtained. The extraction of AOP image features in later experiments validated the effectiveness of the device [62]. The schematic of the polarimeter is shown in Figure 5.

In recent years, the high-temporal-resolution Mueller matrix polarimeters based on liquid crystal modulation have gained increasing attention in real-time measurement fields due to their advantages, including the absence of mechanical moving parts, high temporal resolution, and stable polarization light output [64]. In 2022, Sakamoto et al. [65] used a liquid crystal polarization grating (LCPG) as an amplitude-splitting camera and developed a polarization detection imaging system. This system provides both polarized light illumination and complete Stokes parameter imaging functionality. Compared to the use of rotating polarizers, this system does not require moving mechanical parts to obtain the full-Stokes parameter image of the object of interest. Additionally, unlike the methods based on micro-polarizer arrays discussed later, the absence of crosstalk between pixels caused by diffraction offers the advantage of a superior extinction ratio. The nematic liquid crystal variable retarder (NLCVR) modulates the phase delay by changing the effective birefringence of nematic liquid crystal materials, playing a significant role in retarder fabrication [66]. The ferroelectric liquid crystal phase retarder (FLCPR) has been widely applied in polarization measurements due to its fast, stable modulation characteristics and high resolution. In 2017, Q. Liu et al. [29] used two FLCs and an interferometer to design a time-domain Fourier-transform imaging spectropolarimeter for acquiring spatial, spectral, and polarization information. By rapidly switching the fast axis of the FLC without mechanical movement, the polarimetric state analyzer is able to quickly modulate the full set of Stokes parameters, as shown in the schematic in Figure 6a. In 2020, S. Zhang et al. [67] sandwiched a standard FLCPR between two half-wave plates with parallel optical axes to form an AFLCPR, as shown in Figure 6b. Using a self-developed high-speed Stokes polarimeter, they accurately characterized the AFLCPR, demonstrating that the optical parameters of the ferroelectric liquid crystal retarder depend on the driving voltage. Subsequently, a Mueller matrix polarimeter with a dual ferroelectric liquid crystal retarder was designed based on the self-developed Stokes polarimeter, as shown in the schematic in Figure 6c. The photoelastic modulator (PEM) has garnered increasing attention in recent years due to its high driving voltage, resistance to temperature-induced polarization characteristics, faster phase delay switching speed, and high sensitivity to polarization measurements [14,68]. In 2013, T.Y. Yang et al. [69] used two PEMs and a CCD to achieve fast Stokes vector intensity imaging. To overcome the frequency drift of PEMs, a programmable gate array (PGA) was used to synchronize the PEM with the CCD, resulting in higher image quality. Oriol et al. [70] developed a new type of Mueller matrix polarimeter using four PEMs to simultaneously acquire the 16 elements of the Mueller matrix based on two PEM polarimeters. In 2020, C.Y. Han et al. [71] proposed a method using illumination synchronization to address the issue of the PEM’s available frequency being much higher than the CCD frame transmission rate, achieving dual-wavelength full-Stokes vector coefficient imaging, as shown in Figure 6d.

Electrically controlled phase delay, which provides multiple polarization states’ information, offers advantages over rotating polarizers, such as a faster imaging speed, more stable systems, and the ability to detect dynamic objects without rotating components. However, current advanced electro-optical delay technologies are primarily used in retarder fabrication and laser polarization state detection, with fewer applications in the detection of skylight polarization states for navigation and positioning. In 2023, Vasiliki et al. [72] combined the PEM with a linear polarizer to develop a skylight detection polarimeter. By detecting the polarization of transmitted skylight and observing the dichroic extinction phenomenon, they discovered that dust particles in the atmosphere may exhibit preferential orientation, demonstrating the feasibility of electro-optical delay technology in skylight detection, as shown in the schematic in Figure 6e. This work provides inspiration for the design of future time-sequenced imaging polarimeters.

### 3.2. Channel-Division Polarimeter

Compared to time-division imaging polarimeters, the simultaneous imaging channel polarimeter can capture multiple polarization state images within a single exposure time. Due to its fast detection speed, no moving parts are required, and due to its better stability, it has found widespread applications in polarization detection [73,74]. As previously mentioned, a multi-channel polarimeter refers to a system that uses multiple sensors or different regions of a single sensor to capture images. The implementation can be broadly classified into two approaches: imaging on multiple sensors and imaging on a single sensor. Imaging on multiple sensors can be further divided into two approaches: one that uses optical devices such as beam-splitting prisms or polarizing beam splitters (PBSs) to divide the incident light into multiple beams, with each pass using a different polarizer to obtain multiple polarization state intensity images [75], and another that employs multiple polarization imaging sensors to apply varying degrees of polarization to a single incident light and capture the resulting images [76]. On the other hand, depending on the implementation principle, imaging on a single sensor can be achieved by using techniques like delay plate arrays, lens-focusing effects [35,36], and metasurfaces [77,78].

#### 3.2.1. Design of Multi-Sensor Channel-Division Polarimeter

The amplitude division polarimeter based on multiple independent camera systems can simultaneously capture images with different polarization state intensities from multiple channels, This method is the simplest and most direct way to obtain the required Stokes parameters in the spatial domain [17]. In 2002, Horvath et al. [79] designed an all-sky imaging polarimeter that utilized three lenses and three cameras. By placing polarization filters with angles of 0°, 60°, and 120° in the filter holders of the three fisheye lenses, they achieved the acquisition of polarization state information for the entire sky. A schematic diagram of the device is shown in Figure 7a. In 2014, DB Wang et al. [30] developed a real-time polarization detection polarimeter based on a three-channel camera system. The polarimeter consists of three cameras arranged in a triangular configuration, combining the advantages of both point-source sensors and imaging sensors. Without lenses, it functions as a point-source sensor to measure the polarization information of a single point; With lenses, it operates as an imaging sensor to measure the polarization patterns of an area. To simplify the calculation process, the polarization directions of the three polarization filters are set at 0°, 45°, and 90° relative to the sensor’s reference direction, as shown in Figure 7b. In 2016, J. Tang et al. [80] installed polarization filters in front of the CCD to employ a three-channel polarimeter and establish an all-sky polarization imaging system. This system, combined with a compass information measurement algorithm based on all-sky polarization light distribution images, obtained high-quality polarization images, as shown in Figure 7d. In the same year, C. Fang et al. [25] designed a multi-sensor system consisting of four measurement units, each with polarization angles of 0°, 45°, 90°, and 135°, as shown in Figure 7c. and successfully integrated it with an inertial measurement unit to form a multi-sensor navigation system for urban navigation in 2018 [81].Each measurement unit of this multi-sensor system consists of a CCD camera that equipped with a wide-angle lens and an embedded linear polarization filter. However, due to the lack of a universal calibration algorithm for amplitude division polarimeters [82], and the spatial registration issue is quite complex, which make it difficult for such systems to be widely applied. For this problem, in 2024, X. Qin et al. [83] proposed a step-by-step calibration method based on Singular-Value Decomposition (SVD) and linear fitting for non-uniform grayscale responses of polarizer installation errors, which effectively reduces error accumulation and lowers the complexity of traditional calibration algorithms. The imaging polarimeter structure they employed is shown in Figure 7e.

As early as the 1990s, Azzma utilized amplitude division devices such as beam splitters, PBSs (polarizing beam splitters), and even optical fiber diffraction to divide the incident light into multiple beams with different polarization states [84]. Depending on the number of optical elements used, different dimensional Stokes parameters can be measured [45]. In order to determine the complete Stokes parameters, many polarization imaging polarimeters that use beam-splitting prisms will use three or more optical devices [85,86]. In 2009, Pezzaniti et al. [87] combined an 80/20 partial polarization beam splitter, two 50/50 polarization beam splitters, a 1/4 and a 1/2 waveplate to form a beam splitter block, realizing relatively precise orthogonal polarization imaging, as shown in Figure 8a. In the same year, Fujita et al. [88]. employed a non-polarizing beam splitter placed near the reimaged pupil to separate the incident beam into transmitted and reflected beams. Then, they used two separate Wollaston prisms and waveplates, which divided the incident beam into two orthogonal beams to simultaneously capture four polarization images. The schematic of their setup is shown in Figure 8c. Notably, the orthogonal error between the two sets of mutually orthogonal polarizers is one of the error sources in the sensor, which is why PBS is often used as a replacement for polarizers in other polarization imaging configurations for navigation purposes [89]. For example, in 2018, J. Yang et al. [90]. used PBS as a sensor polarizer to design a polarization navigation polarimeter based on a polarizing beam splitter, as shown in Figure 8b. Moreover, considering the optical path coupling and inconsistent extinction ratios between the transmitted and reflected beams, J. Yang introduced a coupling coefficient to eliminate orthogonal errors and established a new sensor model. Notwithstanding the considerable benefits of polarization image polarimeters utilizing PBS, their substantial dimensions and the complexities associated with the accurate calibration of optical components have resulted in problems for practical applications [91,92].

In summary, the simplest and most direct method to obtain the required Stokes parameters in the design of multi-sensor channel-division polarimetric detection units is to use an amplitude division polarimeter based on multiple independent camera systems. However, the challenges it must overcome include optical response discrepancies and spatial registration issues between multiple polarimetric systems. On the other hand, the split-channel polarimeter based on spectral devices, which utilizes a polarizing beam splitter (PBS) instead of traditional beam-splitting prisms, can eliminate orthogonal errors between two sets of mutually perpendicular polarizers, thereby improving the accuracy of Stokes parameter calculation. However, it should be noted that this method requires a higher level of precision in the optical device structure.

#### 3.2.2. Design of Single Sensor Channel-Division Polarimeter

To obtain dynamic polarization data of the entire daylight sky with a single polarizer, the fundamental theory necessitates the concurrent measurement of all Stokes parameters at the microscopic scale. This necessitates the detection of a minimum of three distinct polarization light intensities, utilizing an array of retardation waveplates or polarizers aligned in various orientations, with imaging conducted in different subregions of the same sensor [93]. In 2017, TAKAFUMI et al. [32] presented a polarization imaging camera with a polarization detection unit constructed from a microwave plate array and linear polarizers, produced by femtosecond direct-write technology. The imaging mechanism of the system segments the waveplate array into many units, each comprising four waveplate regions, as shown in Figure 9a. Each 2 × 2 waveplate region possesses distinct retardation, and when integrated with the linear polarizer, it facilitates the imaging of four polarization intensities. In 2023, J. Xue et al. [94] created a bionic polarization compound-eye system modelled after the architecture of insect compound eyes. This system comprises a nine-hole polarizing film, nine small ocular lenses, a micro-surface optical fiber panel, and a large-format CMOS camera. The polarizing film array is positioned in front of the compound-eye camera at 0° and 45° angles, as shown in Figure 9b. Using the CMOS vertical axis as the reference direction, the polarizer’s orientation at the aperture is ideally set at angles (θ) of 0°, 45°, 90°, and 135°. This design creates a four-aperture linear polarization imaging pattern in the central area of the field of view, facilitating concurrent recording of Stokes vector data within a single CMOS sensor.

Besides employing conventional polarizers, the projection and focusing capabilities of lenses can be used to provide polarized light imaging in various subregions of the sensor [95]. This approach’s value is in the elimination of superfluous beam-splitting components, resulting in an optical system that is stable, straightforward, compact, and conducive to data processing. Nonetheless, the disadvantage is the reduction of spatial resolution. In 2015, WJ Zhang et al. [36] incorporated polarizers orientated at 0°, 60°, and 120° onto the primary lens of the camera. A micro-lens array was subsequently positioned in front of the imaging sensor, dividing each sub-image beneath the array into three zones, each aligned with a distinct polarization direction. Figure 9c illustrates the schematic of the polarization unit and the polarization camera. In 2017, Sturzl et al. [35] utilized the polarization camera setup depicted in Figure 9d, which comprised an S-type lens, three polarizers with varying orientations, a plano-concave lens, and a biconvex lens. This arrangement guarantees that the primary light ray traversing the camera-lens node remains parallel between the biconvex lens and the plano-concave lens. The orientation at the aperture has optimal angles θ of 0°, 45°, 90°, and 135°. This design creates a four-aperture linear polarization imaging pattern in the middle area of the field of view, facilitating simultaneous acquisition of Stokes vector data within a single CMOS sensor.

The critical limitation in the above-mentioned polarization detection units based on metallic wire grids and filters is that they cannot achieve polarization multiplexing because their fundamental energy utilization limit is 50%, and there are pixel-level registration errors. To address these issues, in 2018, Faraon et al. [96] proposed a dielectric metasurface-based polarization imaging scheme by utilizing polarization differentiation and focusing. They organized the polarization unit in a superpixel configuration comprising six image sensor pixels, employing three independent polarization bases (0/90°, 45/135°, and RHCP/LHCP) to measure full-Stokes parameters for each superpixel, as shown in Figure 10a. Finally, they achieved over 60% energy utilization efficiency in the spectral range centered at 850 nm with 10 nm full width at half maximum. This enables the metasurface-based polarization detection unit system to perform well even in weak light environments, and thus it has been widely applied in Stokes imaging polarimeters in recent years. For visible light detection, in 2019, Capasso et al. [97] designed a novel diffraction grating device based on metasurface matrix Fourier optics which can act as a polarizer in any chosen set of polarization states, enabling parallel detection of any set of polarizations on a single optical element and achieving full-Stokes detection, as shown in Figure 10b. Furthermore, they achieved the integration of metasurfaces with a camera to develop a snapshot full-Stokes polarization imaging system that is applicable to remote sensing and even autonomous vehicle technologies. In 2022, T. Xu et al. [98]. proposed a full-Stokes vector polarimeter using all-dielectric metasurfaces. By efficiently splitting three pairs of orthogonal input polarization states (0/90°, 45/135°, and RHCP/LHCP) and simultaneously focusing them on different positions on the sensor plane, as shown in Figure 10c, this configuration enables comprehensive characterization of incident light polarization states while maintaining high energy throughput through measuring six polarized intensity distributions. In 2021, JY Liu et al. [77] used a dual-focus superlens array to form a bionic compound-eye metasurface (BCEM) as the polarization unit. In this system, every three dual-focus superlenses form a bionic sub-eye, each of which decomposes the incident light into two pairs of linearly polarized light and one pair of circularly polarized light, as shown in Figure 10e. Using these light intensities, the Stokes parameters were reconstructed to achieve large-field-of-view full-Stokes polarization imaging. Furthermore, in 2022, JY Liu used a dielectric metasurface to decompose incident light into six polarization components, enabling independent control of orthogonal circularly polarized light. This metasurface can be used alone as a full-Stokes polarization detector or integrated with a standard CMOS to create a high-efficiency polarization imaging CMOS device [78], which will greatly promote the development of daylight polarization detection and navigation applications, as shown in Figure 10d.

We provide three approaches for the construction of a single sensor, channel-division polarimetric detection unit: the application of conventional polarization devices, the focusing capabilities of lenses, and the characteristics of metasurfaces. Conventional polarization devices may still be utilized through the application of retardation plates or linear polarizers; however, distinct polarization treatments are necessary for various sections of the device. The split-channel polarimeter, employing the focusing properties of lenses, provides a straightforward and stable hardware configuration, facilitates easy construction, and matches the dimensions of a standard camera, so enhancing its portability. Furthermore, altering the lens enables the modification of its field of view and angular resolution to accommodate various application contexts. In contrast to filter-based polarizing devices, metasurfaces divide and concentrate incident light, producing a focused spot intensity far exceeding that of the incident light. This attribute renders metasurface-based systems more adept at polarimetric detection in low-light conditions, such as nocturnal settings, offering significant insights for the application of polarized light in navigation during nights.

### 3.3. Division-of-Focal-Plane Polarimeter

Compared to polarization instruments that use imaging across different sensors, the DOFP polarization image polarimeter is based on nano-fabrication technology, allowing for the simultaneous acquisition of four different polarization directions’ daylight information with a single camera and a single sensor, making the measurement of polarization information more straightforward. Unlike polarization imaging systems on the same sensor that rely on principles such as retardation plate arrays, the focusing effect of lenses, and the properties of metasurfaces, the structure and polarization imaging principle of a DOFP polarimeter are more consistent. A typical DOFP polarimeter’s polarization imaging unit consists of a microlens array, a pixelated micro-polarizer array, and an imaging sensor. The micro-polarizer array is divided into multiple 2 × 2-pixel polarization units, each containing four polarizers with different polarization directions, enabling the simultaneous acquisition of four polarization intensities. Subsequently, interpolation, convolution, and other post-processing algorithms are used to derive the wide-field polarization distribution [99]. Based on the fabrication process of the polarization imaging unit, DOFP polarimeters can be classified into non-integrated and integrated designs. Non-integrated DOFP polarimeters suffer from issues such as errors in the installation angle of polarization units and pixel response inconsistencies, often requiring calibration modeling during design. In contrast, integrated DOFP polarimeters effectively reduce optical crosstalk and improve polarization extinction ratios, making them the future mainstream direction for imaging-based biomimetic polarimeters.

#### 3.3.1. The Micro-Polarizer Array Front-Mounted Polarimeter

A typical DOFP polarimeter places the micro-polarizer array (MPA) in front of the focal plane array, with Polaris Sensor Technologies’ Pyxis imager, introduced in 2016, serving as a representative example [100]. In similar works, the micro-polarizers are manufactured individually and then combined with the focal plane array. These DOFP polarimeters are compact, capable of simultaneous imaging, and suitable for dynamic scenes, thus making them widely adopted in most research requiring polarization cameras. Examples include navigation and orientation using nocturnal light at night [101], acquiring daylight polarization mode information [102], improving sensor orientation accuracy under non-ideal conditions (e.g., cloudy skies and sensor tilt) [103,104], and vehicle navigation when combined with other control systems [105,106]. Some of the aforementioned polarization detection unit systems are shown in Figure 11. In 2017, Garcia et al. [107] altered the traditional design approach commonly used in the imaging and vision sensor fields, instead mimicking the visual system of the mantis shrimp to develop a single-chip, low-power, high-resolution color polarization imaging system. This bio-inspired polarization imaging system was realized by combining vertically stacked photodetector arrays with pixelated nanowire filters. These photodetectors can discern three different broadband spectral channels and exhibit polarization sensitivity, as shown in Figure 11d. In 2019, R. Zhang et al. [108] proposed a novel modular DOFP system, which includes a lens, an MPA, a relay lens, and an image sensor. The structure of the polarization imaging unit is compact and easy to operate. The target is imaged onto the MPA via the objective lens, and the intermediate image is projected onto the image sensor via the relay lens, as shown in Figure 11e Due to the flexibility of this system, multiple MPAs and image sensors can be combined through the relay lens for various applications, including navigation and positioning. In 2022, PW Hu et al. [109] assembled an underwater imaging polarization system comprising a fisheye lens, imaging polarimeter, and a level platform with a sealed chamber, as shown in Figure 11f. The DOFP polarimeter was applied to capture underwater images, with the fisheye lens providing a 185° field of view, and the transmission rate of the lens exceeding 50% in the 400–700 nm range, reaching nearly 85% at around 530 nm. The imaging polarimeter was equipped with a Sony IMX250MZR CMOS chip. In the same year, inspired by biological vision, JY Liu et al. [110] implemented the corneal and crystalline cone functions of an insect compound eye (collecting light and focusing it on the retina) by employing an artificial compound eye, and used integrated polarization detectors to simulate the function of the rods and retinal cells (sensing polarized light and analyzing its information), proposing a miniaturized bio-inspired polarization navigation sensor based on an artificial compound eye, as shown in Figure 11g. The micro-lens array on the artificial compound eye detects light intensity in four directions to calculate Stokes parameters. Additionally, to precisely control the field of view (FOV) of individual lenses, aperture diaphragms and field diaphragms were added above and below the micro-lens array. These diaphragms prevent optical crosstalk between adjacent lenses. As a result, opaque walls are not required between the lenses, simplifying the manufacturing process of the micro-lens array.

It is worth noting that conventional DOFP polarization instruments face issues such as polarization unit mounting angle errors, pixel response inconsistencies, and varying instantaneous fields of view across pixels, limiting the accuracy of polarization state measurements. To apply polarization imaging sensors in navigation, in 2017, GL Han et al. [39] considered CCD pixel response errors and installation errors of the pixelated polarizer array, proposing a calibration method based on iterative least squares estimation. In 2020, HN Ren et al. [111] regarding the problem of inconsistent extinction ratios between pixels in camera-based polarization sensors, to improve calibration performance, they introduced an extinction ratio coefficient into the polarization calibration model and proposed a calibration method that considers both AOP and DOP errors to enhance the model’s accuracy. The pixel-level polarization CMOS camera sensor structure used in their study is shown in Figure 12b. Sensor noise is a major factor limiting the measurement accuracy of DOFP polarimeters. In 2021, J Yang et al. [112] demonstrated an error model for DOFP polarimeters based on temporal noise and spatial non-uniformity. Compared to existing models that only analyze temporal noise, they modified the normalization conditions in traditional DOFP polarimeter calibration methods, clarified the selection rules for the coefficient matrix in their proposed model, enabling independent and accurate estimation of errors caused by temporal noise and spatial non-uniformity. In 2023, Y Fan et al. [113] suggest an error model in response to the problems of the four pixels in a superpixel do not measure the same sky region but instead measure with different instantaneous fields of view and vary in azimuth angles, suggest an error model Their error modeling showed that the proposed method effectively reduces polarization measurement errors and improves positioning accuracy based on outdoor experimental results. In 2024, GM Li et al. [114] innovatively applied the Berry model, considering the impact of neutral points, to obtain robust heading information. In the DOFP polarimeter structure they employed, scattered light from the atmosphere enters the array polarization units after passing through a wide-angle lens and a light shield, with each polarization unit consisting of four adjacent pixel channels. Each polarization sensor contains 2048 × 2448 channels, as shown in Figure 12a.

As described in this section, thanks to its high-precision polarization measurement, multi-channel synchronous detection, compact size, real-time control, and other advantages, the micro-polarizer array front-mounted polarimeter is the most widely used polarimeter. However, due to the specific nature of polarization imaging principles, the application of conventional DOFP polarimeters requires the establishment of calibration algorithms to address issues such as installation angle errors of the polarization units, inconsistent pixel responses, and differences in the instantaneous fields of view of the four pixels. On the other hand, the polarimeter with integrated polarizer and pixel array can effectively solve these problems.

#### 3.3.2. The Polarimeter with Integrated Polarizer and Pixel Array

The typical issue with the aforementioned DOFP polarimeters is that, for DOFP polarization sensors, the micro-polarizer array should be placed as close as possible to the image sensor pixels. This means that separately manufacturing and assembling micro-polarizing filters with image sensors will introduce optical crosstalk. However, with the advancement of ultraviolet nanoimprint lithography (UV-NIL) and electron-beam lithography (EBL) technologies, integrating the polarization array with the sensor pixel array has become possible [115]. This greatly reduces the crosstalk of incident light between different pixels [43]. In 2022, CL Guan [116] et al. proposed a flexible UV-NIL-based integration process that enables the integration of micro-polarizer arrays with CMOS image sensors while remaining compatible with standard semiconductor assembly processes. The proposed integrated UV-NIL-based integration process offers excellent manufacturing compatibility, enabling the production of micro-polarizer arrays on any CMOS image sensor without altering the CMOS fabrication process. By replacing molds with different nanoscale patterns, micro-polarizer arrays with different optical properties can be produced, offering excellent adaptability in various application scenarios. The designed polarization sensor and polarization detection unit are systematically illustrated in Figure 13. In the same year, Z Liu et al. [117] combined the characteristics of UV-NIL lithography and other types of lithography to propose a nanoimprint lithography (NIPL) process, which can simultaneously create cross-scale structures with both nanoscale and microscale patterns in one step. By utilizing the NIPL process, they integrated image chips with nano-gratings to create polarization chips, as shown in Figure 14. The polarization navigation sensor they developed achieved a polarization angle measurement error within ±0.2°. In 2024, M. Li et al. [118] utilized nanoimprint lithography to fabricate a composite model on a chip, consisting of a metal–dielectric–metal nanowire grating on top and an anti-reflection coating on the bottom. The model parameters related to polarization capabilities were carefully optimized. The resulting polarization white light exhibits an extinction ratio of up to 40 dB in the wavelength range of 450–750 nm, showing promising potential for applications in polarimetric detection of daylight. The advancements in integrated technology have also been applied in the development of novel materials. In 2023, YY Fan et al. [119] employed nanoimprint lithography and nano-transfer printing technologies to fabricate large-area, ultra-compact 320 × 320 pixelated aluminum-wire-grid full-Stokes metamaterials. The aluminum-wire grid was designed to function as both linear polarization filters and circular polarization filters in the visible and near-infrared spectrum. As a result, the pixelated metamaterial integrated the functionalities of both linear and circular polarization filters, enabling full-Stokes polarization imaging. The measurement errors of the Stokes parameters S0, S1, S2, and S3 were within 8.77%, 12.58%, 14.04%, and 25.96%, respectively. By integrating these technologies, it is expected to achieve full-Stokes polarization imaging with a low extinction ratio.

## 4. Summary and Prospects

Certain species leverage their special eye structures and sensitivity to polarization information to extract directional cues from celestial polarization patterns. The bionic polarization imaging technology inspired by this biological capability has become a focal research area in the fields of bionics and navigation positioning. The bionic polarization imaging system can be classified into three categories according to the configuration of the polarization detection unit: time division, channel division, and focal-plane division. Despite the variation in these system structures, their design concepts are fundamentally consistent, converting incident light with unknown polarization states into well-defined polarized light to extract multiple linear polarization parameters by utilizing the Stokes parameters method. In this paper, we present a comprehensive review of current bionic polarization imaging systems, systematically examining their architectural distinctions and comparative advantages. Currently, the conventional bio-inspired polarization imaging technologies exhibit inherent limitations that can be categorized as follows: large in volume, restricted performance in dynamic scenarios, absence of standardized calibration error algorithms, limited spatial resolution, and the fabrication challenges arising from optically intricate system designs. For instance, the traditional time-division polarization instruments necessitate multiple polarizer rotations to acquire intensity data across varying polarization orientations, thus prolonging single measurement cycles and rendering them not applicable for capturing rapidly evolving dynamic scenarios. Future research priorities should focus on resolving dynamic adaptability challenges in operational environments. Multi-sensor channel-division polarimeters are primarily constrained by optical response inconsistencies, spatial registration discrepancies, and temporal synchronization mismatches across polarization subsystems. A critical challenge lies in establishing universal calibration algorithms for future multi-polarimeter systems. Single-sensor channel-division polarimeters utilizing conventional polarizers eliminate cross-system calibration complexities, but they introduce pixel-level non-uniformity calibration challenges that preserve calibration intricacy. Simultaneously, pixel resource partitioning substantially reduces spatial utilization efficiency. Division-of-focal-plane polarimeters currently dominate optical polarization research; however, the problem is that, in traditional non-integrated fabrication of polarizer arrays and sensor arrays, the instantaneous FOVs obtained by individual pixels differ. Furthermore, micrometer-level alignment errors in polarizer placement near focal planes may induce optical path deviations or image plane mismatches, severely affecting the accuracy of polarization parameter calculation and restricting the measurement precision of polarization states. Although existing bionic polarization imaging systems have achieved notable advancements and found practical applications in navigation and positioning, the aforementioned challenges still pose significant obstacles to the technology’s further development.

The design of polarization detection unit structures indicates that the evolution of polarization information acquisition and detection technology is characterized by miniaturization, integration, and comprehensiveness. Thus, technologies including electric control delay imaging, innovative metasurface-based methods, and full-Stokes imaging with improved grating materials warrant future consideration. Research has shown that the polarization angles detected by the electrically controlled delay devices in daylight polarization state detection exhibit excellent similarity to the Rayleigh model [63]. This aligns with the scientific principles of full autonomy, integration, and real-time technology in modern navigation and positioning. Metasurfaces can be used alone as full-Stokes polarization detectors or integrated with standard CMOS sensors to create high-efficiency polarization imaging CMOS devices. Furthermore, their light-focusing capability surpasses that of traditional polarizer-based filters, making metasurface technology an advanced and indispensable tool in the design of future polarization instruments for navigation applications. Current research on polarization-multiplexing metasurfaces demonstrates steady progress globally. In recent years, inspired by insect vision systems, there are some research advancements have emerged in bio-inspired polarized light-field cameras that synergistically integrate metasurface material properties with biomimetic compound-eye architectures [120]. Future investigations should prioritize addressing chromatic aberrations induced by material dispersion and waveguide effects, while advancing practical implementations in wide-field operational environments [78,121]. In recent years, integrated DOFP polarization instruments have been developed where the polarizer array and sensor pixel array are fabricated together using UV-NIL and EBL technologies, overcoming the issue of crosstalk between incident light from different pixels. Additionally, new grating materials and polarization detection unit structures, fabricated by using advanced nanoimprint lithography and nano-transfer printing technologies, show promise for applications in daylight polarization detection, enabling full-Stokes polarization imaging with a low extinction ratio and thereby improving polarization detection accuracy [111].

The continuous advancement of miniaturized integrated polarization detection units suggests that these novel polarization detection technologies are expected to integrate with bionic polarization imaging systems, addressing technical limitations of existing systems, improving accuracy, and expanding application scenarios, thus becoming the next breakthrough in the field. Ultimately, they will become the next breakthrough in the bio-inspired polarization imaging field, warranting the focus of researchers.

## Figures and Tables

**Figure 1 sensors-25-04069-f001:**
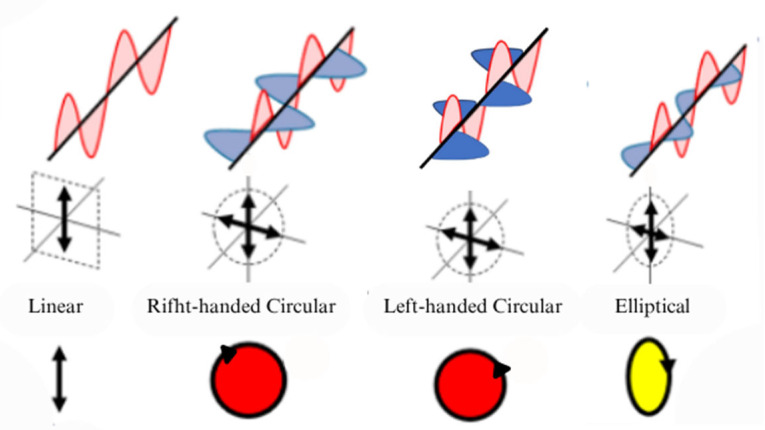
Various types of polarized light [17]. The arrow indicates the rotation direction of the electric field vector along the propagation path.

**Figure 2 sensors-25-04069-f002:**
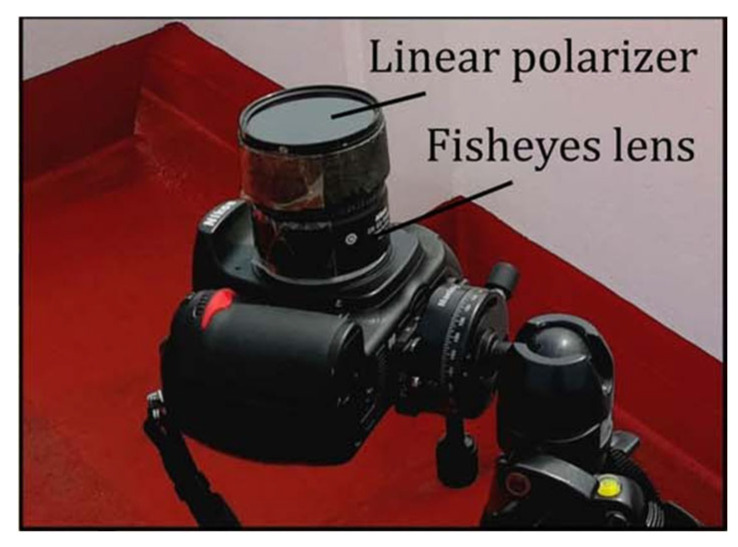
The time-sharing polarimeter used by L Guan [44].

**Figure 3 sensors-25-04069-f003:**
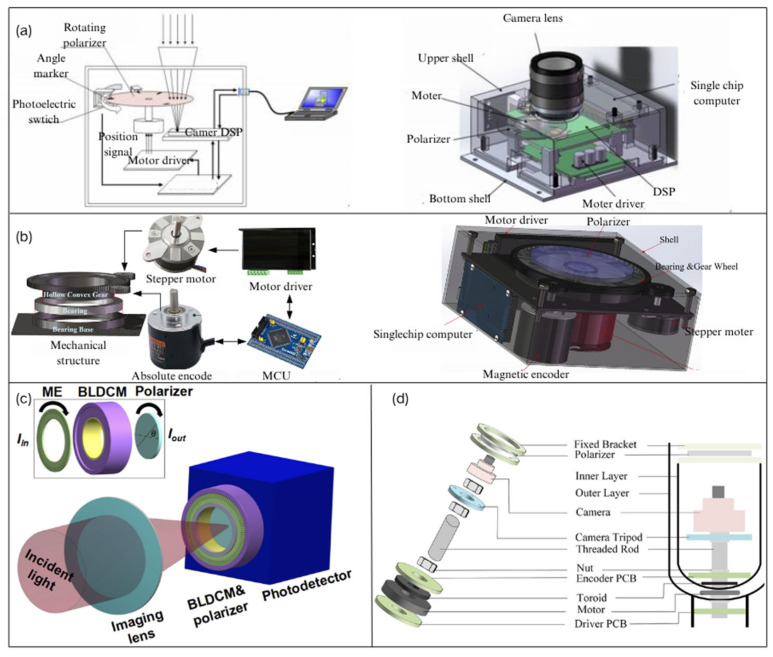
(**a**) Schematic diagram of an improved time-division polarimeter using a rapidly rotating polarizer. The left image shows the overall structure of the polarization imaging system, while the right image presents the design of the polarization camera [53]. (**b**) The PSC-004 time-division polarimeter used by F. Kong. The left image illustrates the automatic control system of the PSC-004’s mechanical polarization rotation device, and the right image shows the structural design of the PSC-004 [56]. (**c**) Schematic diagram of a time-integrating polarimeter with a continuously rotating polarizer used by N. Gu [54]. (**d**) Time-division polarimeter used by X. Lu [57].

**Figure 4 sensors-25-04069-f004:**
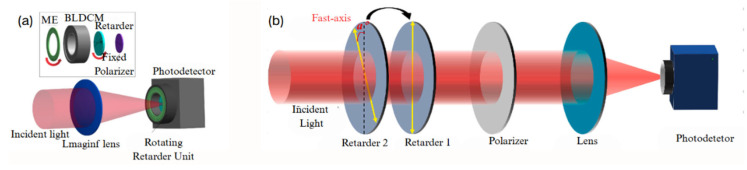
(**a**) The BLDCM that is designed by substituting the spinning polarizer with a rotating retarder [59]. (**b**) The dual-delay time-division polarimeter employed by JP Yang [48].

**Figure 5 sensors-25-04069-f005:**
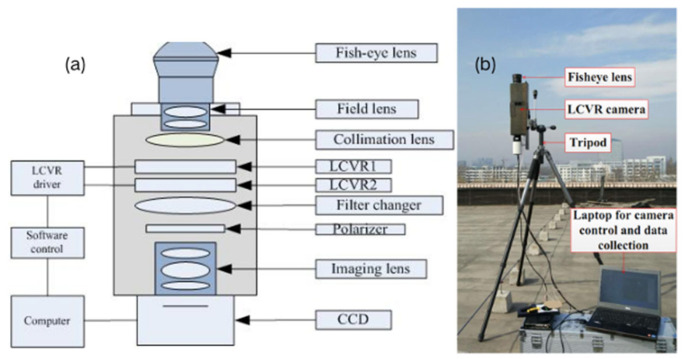
(**a**) The schematic diagram of the ground-based all-sky imaging polarimeter utilizing LCVR technology. (**b**) The schematic representation of the experiment [62].

**Figure 6 sensors-25-04069-f006:**
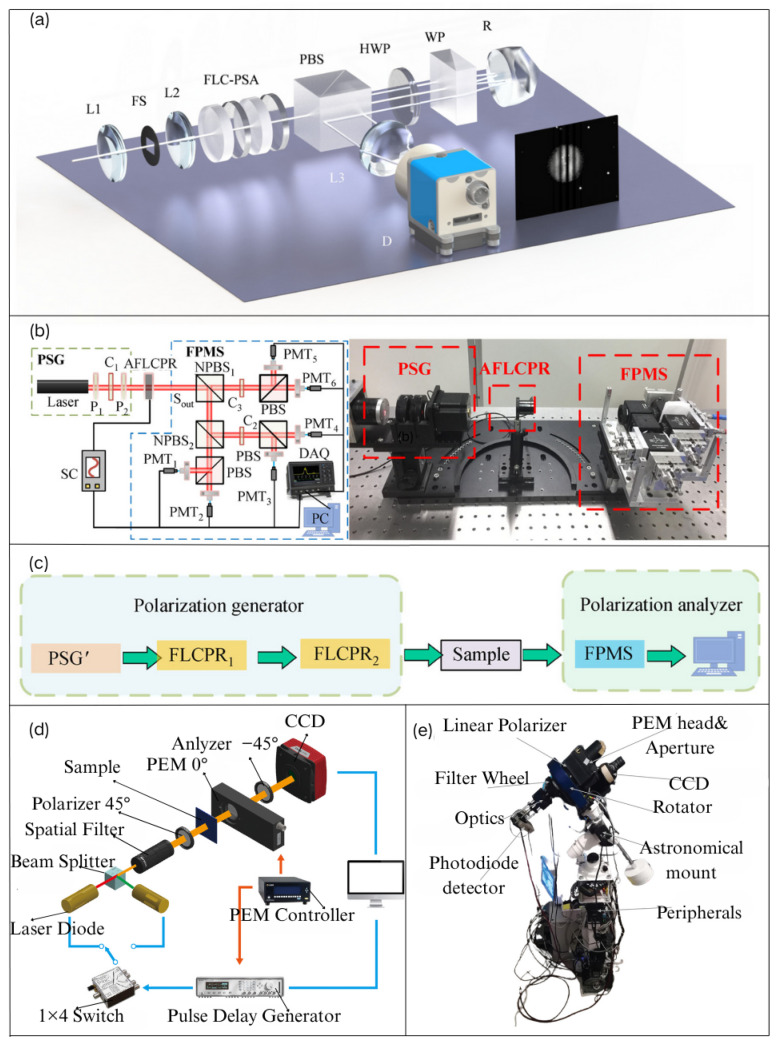
(**a**) Schematic representation of a time-division Fourier-transform imaging spectroscopic polarimeter employing ferroelectric liquid crystal (FLC) and a Wollaston interferometer [29]. (**b**) The experimental schematic for the characterization of AFLCPR [67]. (**c**) The schematic representation of a Mueller-matrix polarimeter using a dual-ferroelectric liquid crystal retarder [67]. (**d**) Optical arrangement of the Stokes polarimetric imaging system employed by CY Han [71]. (**e**) The polarimeter for skylight detection employed by Vasiliki [72].

**Figure 7 sensors-25-04069-f007:**
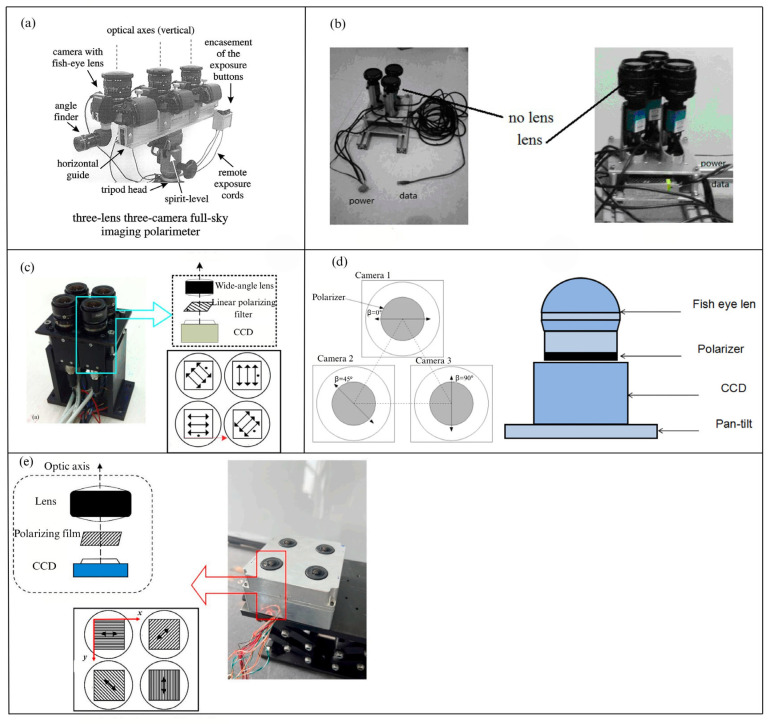
(**a**) The setup of the 3-lens, 3-camera, 180° all-sky field-of-view polarization imaging system used by Horvath [79]. (**b**) The polarimeter combining both point-source and image-based approaches [30]. (**c**) The polarization instrument designed by C. Fang [25]. (**d**) The all-sky polarization system: The left image shows the all-sky polarization imaging system used for polarization detection, while the right image illustrates the polarization imaging system composed of a fisheye lens and CCD [80]. (**e**) A schematic of a bionic polarization navigation system used for capturing daylight polarization patterns. The system includes four optical systems, each consisting of an optical lens assembly, a polarizer, and a CCD detector used to detect daylight polarization patterns [83].

**Figure 8 sensors-25-04069-f008:**
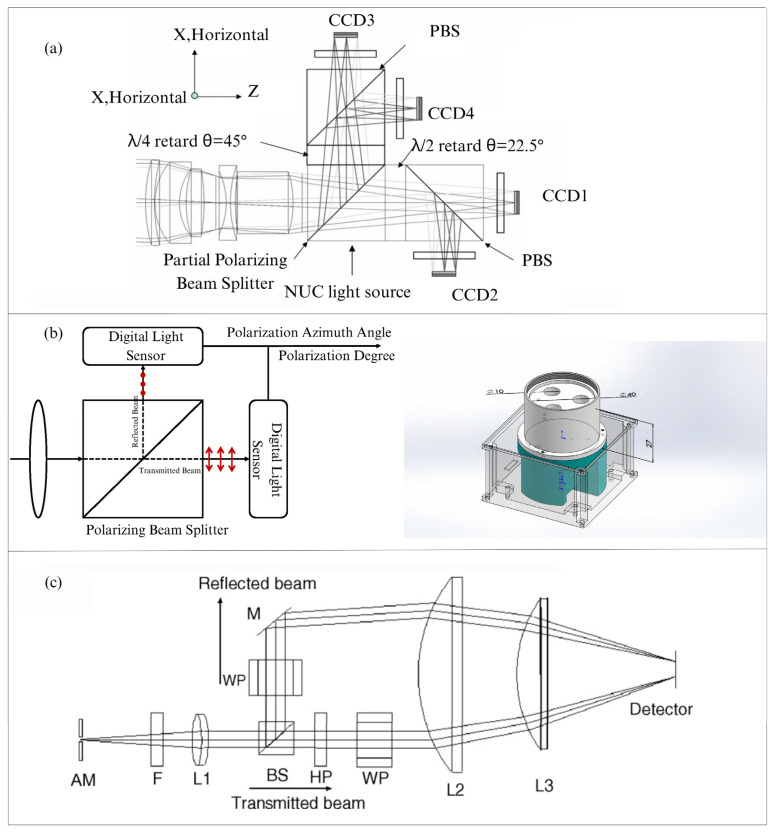
(**a**) The full-Stokes imaging polarimeter used by Pezzaniti [87]. (**b**) The polarimeter utilizing a polarization beam splitter developed by J Yang. The left illustration depicts a schematic of the PBS, while the right illustration presents a schematic of the experimental configuration [90]. (**c**) The polarimeter used by Fujita [88].

**Figure 9 sensors-25-04069-f009:**
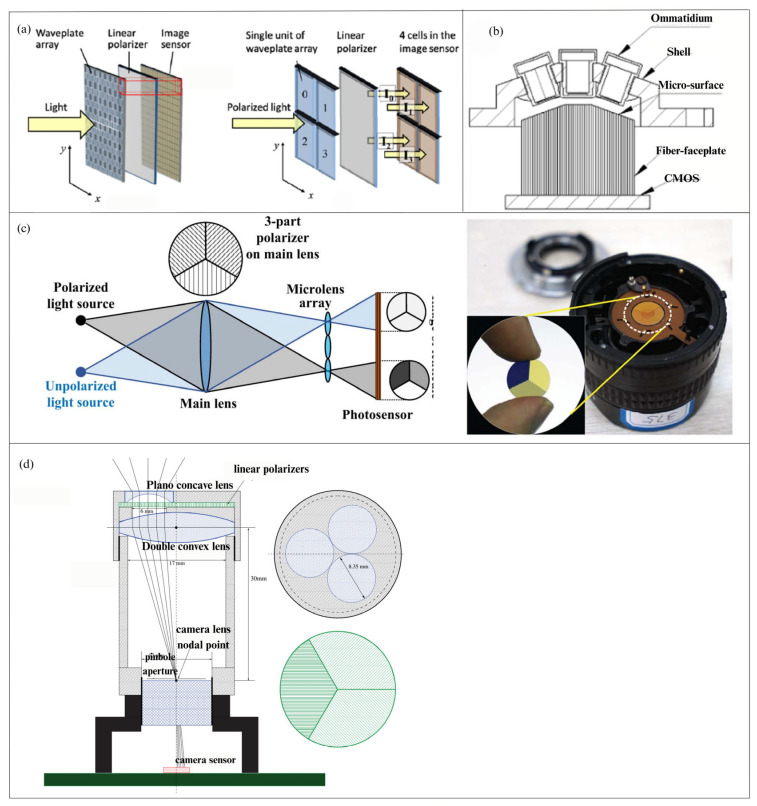
(**a**) The upper image is the schematic diagram of the polarization unit. The lower image shows the structure of the polarization camera, where the relay lens serves to project the image onto the sensor [32]. (**b**) The schematic diagram of a bionic compound-eye camera structure [94]. (**c**) The left image shows the optical path diagram of the imaging system, and the right image displays a wide-angle lens with a linear polarizer triple mirror [36]. (**d**) The imaging system schematic used by Sturzl. Linear polarizers (green) in different orientations are positioned behind three plano-concave lenses. The camera lens is located on the focal plane of a double-convex lens [35].

**Figure 10 sensors-25-04069-f010:**
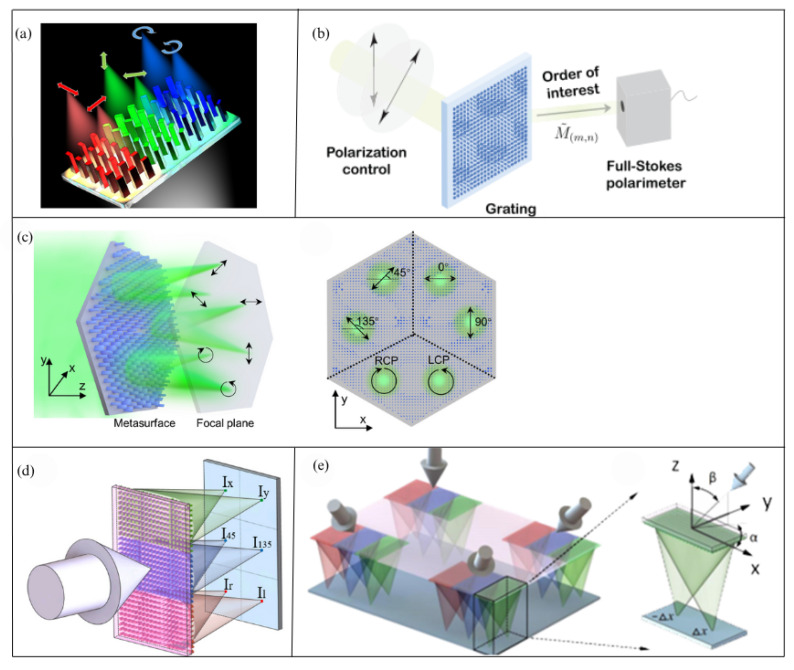
(**a**) Three-dimensional illustration of a superpixel focusing different polarizations to different spots [96]. (**b**) The diffraction grating device based on metasurface matrix Fourier optics [97]. (**c**) The planar metasurface consisting of three polarization beam splitters spatially arranged on a hexagonal pattern proposed by T. Xu [98]. (**d**) The schematic of the polarization detection metasurface [78]. (**e**) The schematic of BCEM and the double-focus superlens in the sub-eye, where α and β represent the azimuth angle and elevation angle of the sub-eye’s optical axis, respectively [77].

**Figure 11 sensors-25-04069-f011:**
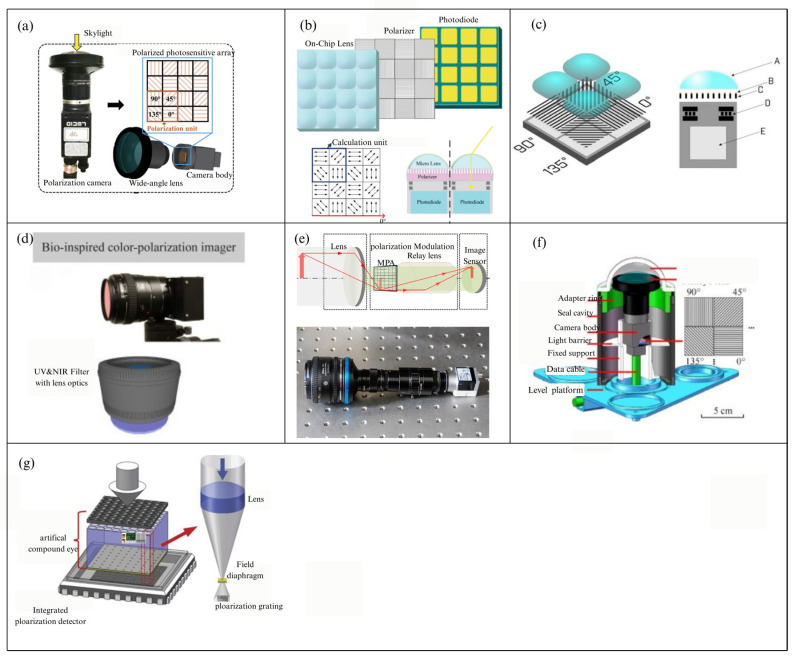
(**a**) The schematic diagram of the DOFP polarization instrument used in navigation and orientation research under sensor tilt conditions [101]. (**b**) The DOFP polarization instrument applied for nighttime navigation using nocturnal light [104]. (**c**) The schematic diagram of the polarization unit of the DOFP polarization instrument applied in combination with an inertial unit for autonomous vehicle navigation [105], The polarization filter (C) of each pixel, coated with an anti-reflection layer (B), is positioned between the lens (A) and the photosensitive photodiode (E). (**d**) Color polarization imaging sensor. It combines vertically stacked photodetectors and pixelated metal nanowires, with the former used for spectral sensitivity and the latter for polarization sensitivity [107]. (**e**) The upper image shows the optical path diagram, and the lower image shows the appearance of the DOFP polarization instrument [108]. (**f**) The 3D interactive section of an underwater imaging polarization sensor [109]. (**g**) A schematic diagram of the polarization imaging unit and measurement optical path of the polarization instrument inspired by the insect compound-eye structure [110].

**Figure 12 sensors-25-04069-f012:**
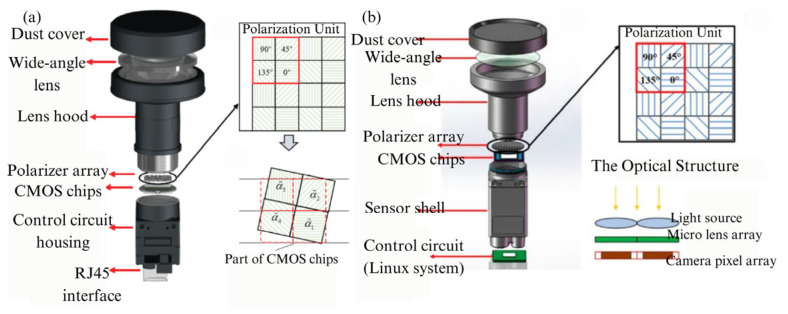
(**a**) Schematic diagram of the imaging polarimeter and error correction unit used by GM Li [114]. (**b**) Schematic diagram of the bionic imaging polarimeter and error correction unit designed by HN Ren [111].

**Figure 13 sensors-25-04069-f013:**
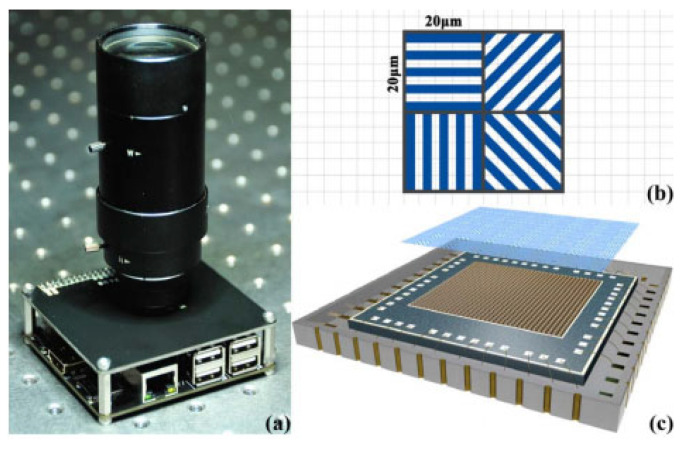
(**a**) Photo of the integrated micro-polarizer array sensor. (**b**) Schematic diagram of the micro-polarizer array. (**c**) The polarization-related detection module fabricated by integrating the micro-polarizer array with CMOS through UV-NIL [116].

**Figure 14 sensors-25-04069-f014:**
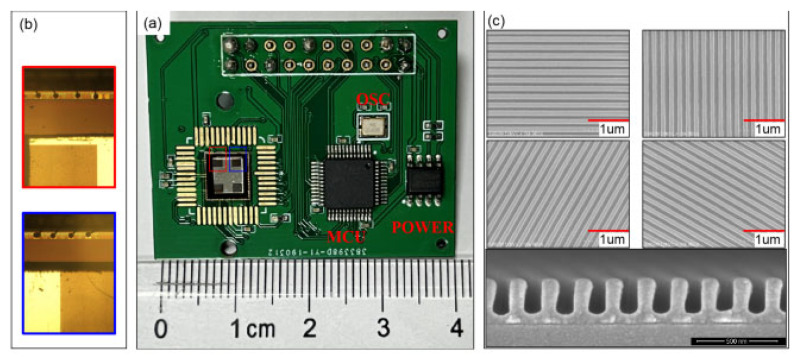
The polarization chip fabricated by integrating nano gratings with image chips by using NIPL technology [117]. (**a**) The photograph of the integrated polarization sensor. (**b**) Partially enlarged detail photographs of the nanogratings and adjacent electrodes on the polarization chip. (**c**) SEM images of multidirectional nanogratings on a silicon test piece fabricated via the same integration process.

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
