# Peer review of "The Structural Types of the Polarization Detection Unit in Imaging Polarimeter Based on the Stokes Parameter Method"

_sensors, 2025, doi:10.3390/s25134069_

Round 1

Reviewer 1 Report

Comments and Suggestions for Authors

1. Figure 1: Missing Circular Polarization Labels
Figure 1 illustrates various polarization states but does not explicitly label left-handed vs. right-handed circular polarization. Please update the figure caption and related text to identify “L-handed circular polarization (LHCP)” and “R-handed circular polarization (RHCP)” clearly.

2. Inconsistent Notation in Equations (4) & (5)
The manuscript refers to “degree of linear polarization (DOLP)” and “degree of circular polarization (DOCP)” in the text, but Equations (4) and (5) use “LDOP” and “CDOP.” These symbols are inconsistent. Choose one notation (e.g. DOLP/DOCP) and apply it uniformly throughout, defining the abbreviations on first use.

3. Incorrect Statement on Number of Parameters (Line 110)
Line 110 states that “DOP and AOP require only three unknowns,” but Equation (3) shows that DOP calculation involves all four Stokes parameters (I, Q, U, V). It appears the authors meant that DOLP (linear polarization only) can be solved with three parameters when V=0. Please correct the wording and clarify that Equation (7) applies only in the absence of circular polarization.

4. Section 2 Is Logically Disorganized
The flow in Section 2 (“Optical Polarization State Detection Principle Based on the Stokes Parameters”) jumps between Stokes theory and measurement methods without clear transitions. I recommend reorganizing into subsections:

  • 2.1 Stokes Vector and Mueller Matrix Overview

  • 2.2 Four-Angle Measurement Method (for full Stokes)

  • 2.3 Three-Angle Measurement Method (for pure linear polarization)
    Add brief introductory sentences for each subsection to guide the reader.

5. Duplicate Figure Numbering
Two separate figures are both labeled “Fig. 1.” Please renumber the second occurrence as “Fig. 2” and update all in‐text references accordingly.

6. Formatting Issue (Lines 360–368)
The text in lines 360–368 appears misformatted (line breaks, indentation, or font styles). Please correct the formatting so that it aligns with the rest of the manuscript.

7. English Language Proofreading
The manuscript would benefit greatly from professional proofreading by a native English speaker to improve grammar, syntax, and overall readability.

8. Figure Layout Optimization
Many of the multi‐panel figures are cluttered and unevenly spaced. Please optimize the layout (e.g. adjust panel sizes, align subfigures, standardize font sizes) for better visual clarity.

Reviewer 2 Report

Comments and Suggestions for Authors

The review article focuses on the structural design of imaging polarimeters, particularly emphasizing systems developed for detecting the polarization state of skylight. The topic is relevant, and the manuscript does attempt to provide a comprehensive overview of the subject. However, in its current form, the article requires major revisions to improve both its readability and accuracy.

Suggestions & Concerns:

  1. Figure Referencing and Layout Issues:
    • There are several instances of incorrect or confusing figure references that significantly hinder the understanding of the content.
      • For example, Figure 2 is incorrectly referred to as Figure 11 on line 243, and also the corresponding Figure legend is also incorrect.
      • The subfigures in Figure 3 (line 285) are not clearly referenced, making interpretation difficult.
      • A reference to Figure 18 is made on line 426, yet no such figure exists. Similar issues are present throughout the manuscript.
    • Additionally, the arrangement of subfigures lacks clarity and logical structure, which makes it challenging to follow the narrative associated with them.
  2. A detailed discussion on Limitations of Existing Systems:
    • The manuscript would benefit from a dedicated section discussing the current limitations and challenges faced by existing polarimetry systems. This could be incorporated into the current Figure 3.4 section or introduced as a standalone discussion.
  3. Missing References to Recent Developments:
    • The authors mention advanced concepts such as simultaneous Stokes vector and Mueller matrix imaging, but fail to cite recent developments in metasurface-based polarimetric systems. Including references to relevant state-of-the-art research in this area would enhance the completeness and relevance of the review.

 In summary, while the topic and intent of the review are appreciated, the manuscript requires significant revisions before it can be considered for publication. The key areas to address include correcting figure references, reorganizing subfigures, improving the text, and adding more comprehensive and recent references.

Reviewer 3 Report

Comments and Suggestions for Authors
  1. paper advantages

The value of the review is outstanding: the article focuses on the structure type of polarization detection unit of Stokes parameter method, and covers time-division, channel-division, and focal-plane division And the performance of various structures is compared horizontally, which has high academic reference value.

The technical classification is reasonable: the technical classification framework in Chapter 2-3 is clear-cut, and the analysis is carried out from the dual perspectives of optical system design and detector design. It is logical and rigorous, and conforms to the research paradigm in the field.

  1. problems and suggestions to be modified

(1) Format normalization

Problem of Title Language: the title of Figure 14 is currently in Chinese (such as "schematic diagram of optical path of polarizing beam splitter prism"), which does not conform to the publication specification of English journals. It is suggested to uniformly modify the description to English and check whether other charts have similar problems.

(2) Section 3.4 redundancy: this part is the summary and outlook of Chapter 3, which is the same as the summary and outlook structure of the full text in Chapter 4. It is suggested to move the outlook part in 3.4 into Chapter 4.

(3) Chapter 4 refers to "bionic polarization imaging technique" rather than "polarization detection unit". It is suggested to modify the language description of the summary part and strengthen the point.

  1. revise expectations and conclusions

This paper has reached the employment standard in terms of the value of the topic and the integrity of the content, but it needs to be revised for the standardization of language expression and format. It is suggested that the author should check the terms of the chart item by item, modify the statements in 3.4 and Chapter 4, and mark the modification position in the revision description. If the above problems are properly solved, this article will have high publishing value.

Round 2

Reviewer 2 Report

Comments and Suggestions for Authors

The manuscript has been improved substantially through the modifications of the figures, correcting the figure references, inclusion of additional discussions on the aspects of limitations associated with the existing approaches. Indeed the manuscript presents a comprehensive view on the existing polarization detection unit systems with emphasizing their designing methods and potential applications. I believe it would be appealing for a broad scientific community and is suitable for publication.